# AI-Powered Telemedicine for Automatic Scoring of Neuromuscular Examinations

**DOI:** 10.3390/bioengineering11090942

**Published:** 2024-09-20

**Authors:** Quentin Lesport, Davis Palmie, Gülşen Öztosun, Henry J. Kaminski, Marc Garbey

**Affiliations:** 1Care Constitution Corp., Newark, DE 19702, USA; quentin.lesport@etudiant.univ-lr.fr (Q.L.);; 2Laboratoire des Sciences de l’Ingénieur pour l’Environnement (LaSIE) UMR-CNRS 7356, University of La Rochelle, 17000 La Rochelle, France; 3Department of Neurology & Rehabilitation Medicine, School of Medicine & Health Sciences, George Washington University, Washington, DC 20037, USA; 4Department of Surgery, School of Medicine and Health Sciences, George Washington University, Washington, DC 20037, USA

**Keywords:** telehealth, telemedicine, myasthenia gravis, ptosis, diplopia, neurological disease, deep learning, computer vision, eye-tracking, clinical trial

## Abstract

Telemedicine is now being used more frequently to evaluate patients with myasthenia gravis (MG). Assessing this condition involves clinical outcome measures, such as the standardized MG-ADL scale or the more complex MG-CE score obtained during clinical exams. However, human subjectivity limits the reliability of these examinations. We propose a set of AI-powered digital tools to improve scoring efficiency and quality using computer vision, deep learning, and natural language processing. This paper focuses on automating a standard telemedicine video by segmenting it into clips corresponding to the MG-CE assessment. This AI-powered solution offers a quantitative assessment of neurological deficits, improving upon subjective evaluations prone to examiner variability. It has the potential to enhance efficiency, patient participation in MG clinical trials, and broader applicability to various neurological diseases.

## 1. Introduction

Telemedicine (TM) is an emerging tool for monitoring patients with neuromuscular disorders and has significant potential to improve clinical care [1,2], especially in patients with favorable impressions of telehealth during the COVID-19 pandemic [3,4]. There is great promise in taking advantage of the video environment to provide remote alternatives for physiological testing and disability assessment [2].

Myasthenia gravis (MG) is a rare neuromuscular disorder which manifests with a broad spectrum of weakness of the voluntary skeletal muscle [5]. It occurs when the body’s immune system mistakenly attacks the neuromuscular junction, which is where nerve cells communicate with muscle cells. This attack disrupts the transmission of nerve signals to the muscles, leading to muscle weakness. The severity of manifestations varies over time and among patients, with some individuals experiencing life-threatening weakness requiring hospitalization.

Telehealth is particularly well-suited for the management of patients with MG. The potential to perform frequent remote monitoring of patients would be expected to enhance routine medical care and potentially improve the efficiency of clinical trials [6,7,8]. Currently, existing practices in telemedicine for the assessment of MG heavily depend on the experience of the neurologist. The limitations of these telemedicine sessions are shaped by examiner availability (e.g., availability of providers in rural settings), changes in providers due to patients having to relocate, and variations in assessment scores based on the subjective nature of the assessment administered by the provider. As there is no objective comparability established in scoring compared to previous visits, current practices entail variations in the examiner’s adherence to the prompts while administering the neuromuscular exam and technical issues regarding patient video, microphone, and internet connection. For the purpose of remote evaluation of MG patients, the Myasthenia Gravis Core Exam (MG-CE) [9] and Myasthenia Gravies Activities of Daily Living (MG-ADL) were conducted via telemedicine to evaluate the symptoms and functional limitations of the disease.

Clinical trial outcome measures for many neurological diseases are compromised by subjectivity, poor reproducibility across evaluators, and the need for in-person evaluations [10]. We propose a digital solution named “inteleclinic” to provide a quantitative analysis of the neuromuscular examination performed via telemedicine. Our technology offers the potential to reduce costs and increase accuracy through the following logic:(1)The quantitation of neurological deficits is an improvement of the existing outcome measures that are subject to human evaluation with variability among examiners, are at best categorical in nature and not continuous metrics, and are dependent on the availability of examiners. We are also able to capture deficits that are better detected by our technology, which human assessments may potentially miss [11].(2)Since measures can be performed remotely, a central evaluation unit with a limited number of highly trained, research coordinators can perform evaluations in a more uniform fashion [12].(3)Inteleclinic allows more complex data collection in a remote location; therefore, research subjects do not need to travel to clinic sites.(4)The solution has the potential to broaden clinical trial participation to poorly represented groups, including international participation, because of reduced barriers such as excessive travel time, conflicting employment requirements, or the subject’s own disability [10]. We have constructed the computational framework in [11,13] to assess MG-CE scores digitally. To validate our algorithms, we applied them to a large cohort of patients with MG and a diverse control group. The results indicate that our approach yields outcomes that are consistently comparable to those of expert clinicians for the majority of examination metrics under standard clinical conditions [13]. We also determined which parts of the examination were not reliable and revised our understanding of how certain sections of the MG-CE should be interpreted [9].

This paper proposes a further step to make the MG-CE examination run fully automatically by identifying each part of the video examination and segmenting the video. A simple graphic user interface can run the annotated video of the patient evaluation produced by our system in fast mode to allow a certified neurologist to review the analysis and validate the score promptly, therefore, speeding up the process by one order of magnitude. This study is significant in terms of enabling more complex data collection while providing reproducibility of scores and facilitating comparison with previous patient visits in an objective fashion, while providing easy access and shorter time on both patient and provider ends.

## 2. Materials and Methods

### 2.1. Subjects and Video Recording

We recruited participants to undergo standardized examinations via telemedicine to assess their performance on the MG-Core Examination (MG-CE) [1,9]. During these telemedicine sessions, examiners began by asking patients to complete the standard MG-ADL score [8] and follow-up with the MG-CE protocol. A detailed description of the MG-CE can be found in [1,9]. Briefly, the evaluation assessed eight key domains: two ocular, three bulbar, one respiratory, and two limbs. Each domain received a severity score ranging from 0 (no deficit) to 3 (severe deficit). We added a score of 4 to indicate examinations that were unusable for the analysis.

The complete dataset consisted of 102 videos of 51 patients performed by eight board-certified neurologists from five different academic hospitals across the USA. Each subject underwent the MG-CE twice within seven days, except for one patient with a 39-day gap between recordings. Participation required access to a video-enabled laptop or tablet with a stable internet connection. All participants were instructed to be in a well-lit area with a hard-backed chair and enough space to stand up while on the camera.

Following our initial publications using pre-existing MGNet videos of MG patients [6,7], we expanded our dataset to include 24 videos of healthy control subjects. The role of this control group was to clearly separate the quantitative assessment of myasthenia gravis symptoms from asymptomatic. As a matter of fact, the digitalization of MG symptoms provides a continuous metric and no longer a categorical assessment based on the clinician’s judgment [11,14]. Two new examiners were trained to improve protocol adherence and video acquisition quality while still utilizing standard Zoom calls. Control subjects with no self-reported physical limitations and with a score of zero on the MG Activities of Daily Living Outcome Measure were recruited. We specifically sought a control group with diverse ethnic and racial backgrounds, equal gender representation, and ages ranging from 18 years to no upper limit. Controls underwent the MG-CE once. All evaluations were performed by board-certified neurologists, and patient examinations were conducted by neurologists with additional certification in clinical neurophysiology.

All participants provided written informed consent for inclusion in the study. The patient study was approved by the central institutional review board of MGNet at Duke University and the George Washington University Institutional Review Board. This control study was approved by the George Washington University Institutional Review Board.

### 2.2. Overall Algorithm Approach

Our method automatically analyzes telemedicine videos of the MG-ADL assessment followed by the MG-CE to generate patient scores. The process consists of three main steps:

Video Segmentation: The video is automatically segmented into clips corresponding to each individual MG-CE test. This includes separating the preparation time (physician explaining the test), the active testing window (patient performing the test), and the concluding time (physician comforting the patient and transitioning to the next test).

Digital Scoring: Each segmented video clip is analyzed by our digital scoring tool to compute a score based on the MG-CE criteria. This tool, previously described and validated in [11,13,14], focuses solely on scoring the active testing window and is tolerant to variations in the preparation and concluding periods.

Automatic Report Generation: By combining segmentation and scoring, our algorithm automatically generates a patient score report. This report can be directly uploaded to the electronic patient record.

The first 8 questions of the MG-ADL score are explicitly written in the protocol, but examiners do not necessarily follow the protocol precisely nor use the same vocabulary. We identify the ADL questionnaire in the video in order to extract the starting time for the MG-CE with ptosis.

The MG-CE scores are in principle following the order: (1) Ptosis → (2) Diplopia Right → (3) Diplopia Left → (4) Cheek Puff → (5) Tongue-to-Cheek → (6) Count to 50 → (7) Extend Arm → (8) Single Breath Count → (9) Sit-to-Stand—see Table 1.

However, we observed variations in how the examiners conducted the MG-CE in our videos. While the ocular exam consistently starts first, the order of diplopia assessments (right vs. left eye) can differ. Additionally, we identified some variations in the sequence of the remaining tests, with occasional omissions due to patient limitations.

This variability is likely due to human behavior [15]. When tasks exceed three or four, memory lapses and deviations are expected without an enforced electronic checklist. However, clinical examinations often prioritize natural social interactions over rigid adherence to order.

The tests can be set into 4 categories—see Table 1 and Figure 1.

Ocular test to assess the severity of ptosis and diplopia for the patient; this counts for three tests with the patient looking up while keeping their head fixed at the center for 60 s to a target on the ceiling or directing his/her gaze left for 60 s and then right for 60 s. Each test starts with an explanation and, eventually, a demonstration provided by the doctor, followed by the 60 s exercise, and concluded by a short patient and doctor conversation to let the patient relax and be prepared for the next exercise.

A speech test is when the patient is asked to count from 1 to 50, and later count as long as possible on one breath. The first test is intended to evaluate the presence of dysarthria (slurred speech), while the second test evaluates any eventual weakness in breathing.

Cheek puff and tongue-to-cheek push are two additional tests used to assess cheek and tongue muscle weakness.

Finally, bulbar and limb weakness are assessed, respectively, by asking the patient to extend their arms horizontally for as long as two minutes and to sit down and stand up with their arms crossed if feasible.

While the general structure (explanation, demonstration, exercise, conversation) applies to most tests, we observed some variations in the order in which examiners conducted the non-ocular tests (speech, bulbar, limb) due to the factors discussed previously.

Our goal was to automatically detect each part of the examination with just enough accuracy to segment a video clip that could be analyzed by a specialized digital tool to provide a score, as described in our previous work [11,13]. The digital score is then computed solely for this active testing window.

The difficulty in segmenting videos varies depending on the MG-CE test being performed. We adopted a “divide-and-conquer” approach, prioritizing easier tests for segmentation before tackling more challenging ones. This strategy utilizes a combination of the following information channels:

We leverage an AI transcription tool, AssemblyAI [16], to generate a word-by-word transcript of the telemedicine session. This transcript includes timestamps, speaker identification (doctor vs. patient), and word confidence scores. However, due to potential sound quality issues or speech classification difficulties, the transcript may contain errors.

We employ natural language processing (NLP) techniques to identify clinical keywords specific to each MG-CE test within the transcript. Although examiner communication styles may vary, these keywords offer an initial estimate of each video clip’s start and end times. However, this approach alone cannot pinpoint the exact start and end of an active testing window.

This combined approach provides a foundation for segmentation, with subsequent steps refining the boundaries of the active testing window for each test.

### 2.3. General NLP Method

We feed the audio to the AssemblyAI API, which generates a transcript of the video. For each word, we have the start and end timestamps in the audio. To improve keyword search accuracy, we perform some basic text cleaning. We remove punctuation marks from the transcript to simplify keyword matching. We convert all text to lowercase for case-insensitive keyword searches.

Next, we employ NLP techniques to identify keywords indicative of specific MG-CE tests within the transcript. Our approach iterates through the transcript. We search for the first word in the target keyword phrase. If the subsequent words in the transcript match the remaining words in the phrase, we identify a keyword match.

The timestamp corresponding to the first word in the matched phrase becomes the initial estimate of the video clip’s start time. Table A1 in Appendix A provides examples of the keywords used for this purpose.

When searching for a multiple-worded expression, we iterate through the words in the transcript until we match the first word in the expression; then, if the following words match the following words in the expression, we have a match. The output timestamp corresponds to the start time of the first word in the expression. Table A1 in Appendix A provides an example of the dictionary we use below.

ADL score test: All videos start with the standard MG-ADL evaluation with 8 specific questions the patient should answer. The following words should be placed in the transcript in the correct order and time window: (Q1) slurred nasal speech, (Q2) fatigue jaw chewing, (Q3) choking swallow, (Q4) shortness of breath, (Q5) brushing teeth, (Q6) trouble chair toilet, (Q7) double vision, and (Q8) drooping or droopy eyelids. Some sections of the video are easy to detect using NLP. The ADL questionnaire should follow a specific script. Unfortunately, only half of these are effectively used by doctors on average [15]: we use a frequency analysis of words from doctors and patients to maximize the rate of detection of these questions and determine when the last question has been answered.

We can look next to the video clip of the MG-CE score, as in Table 1.

### 2.4. Counting Exercises

Identifying the active test window for both counting exercises, count to 50 and single breath count, is straightforward due to their fixed word sequences. We leverage the transcript to pinpoint the exact start and end times of these exercises by searching for a specific number series. Our algorithm tolerates missing numbers in the transcript due to potential recognition errors.

While patients should ideally not count beyond thirty during the single breath test, strict enforcement may not always occur. Additionally, we analyze the counting rate through word frequency in the transcript. This helps to detect instances where patients may count faster or accelerate their speech, particularly during the one-breath test.

Interestingly, inaudible or poorly recognized numbers toward the end of the single breath count transcript act as an objective metric for determining when the patient can no longer continue counting.

By identifying and discarding the time windows corresponding to the MG-ADL score and counting tests, we narrow the search window and make segmentation of the cheek puff and tongue exercises easier based solely on transcript analysis.

### 2.5. Cheek Puff and Tongue-to-Cheek Exercises

Segmenting cheek puff and tongue exercises leverages the distinct vocabulary used in their instructions, minimizing confusion with other tests. We employ the following approach: we first search the transcript for keywords specific to each exercise’s explanation. Then, we utilize a k-means clustering algorithm to identify two distinct clusters corresponding to the two exercise introductions. The average time within each cluster is used as the estimated start time for the respective exercise segment. If keyword detection fails in these brief exercises (typically lasting only a few seconds), we employ a computer vision-based segmentation method (described in [17]) as an alternative.

This two-pronged approach improves, to some extent, the robustness of segmentation for cheek puff and tongue exercises.

### 2.6. Ocular Exercises

Segmenting the ocular tests (ptosis, diplopia left, diplopia right) is more complex than other tests due to the combination of visual and verbal cues. We leverage a two-pronged approach using both natural language processing (NLP) and computer vision (as illustrated in Figure 2).

We first use a rule-based NLP classification to identify the start and end of the ocular phase within the transcript. Keywords related to the start of ocular exercises (e.g., “look up”) trigger the beginning of the phase, while keywords related to subsequent exercises (e.g., “cheek puff”) mark the end. This narrows down the video segment for further analysis. (Refer to Table A1 in Appendix A for keyword examples).

Within the identified ocular phase, we employ computer vision to segment individual exercises (ptosis, diplopia left, diplopia right) based on gaze direction. We use a Haar cascade classifier (OpenCV [17]) to detect the patient’s face and then fit a 68-point landmark detector (DLib [18]) to locate key facial features, including the eyes. A convolutional neural network (CNN) classifies video frames into four categories: “up”, “left”, “right”, and “neutral”, indicating the patient’s gaze direction. This network is trained on video frames from our dataset, where frames during ptosis exercises are classified as “up,” diplopia exercises as “left” or “right”, and gaps between exercises as “neutral” (refer to [11,18] for details).

We analyze the Convolution Neural Network output over the entire ocular phase. A sliding window calculates the “density” of each gaze class (up, left, right) over a 5-s interval. The exercise segment is identified as the longest contiguous period where a specific gaze class density exceeds 60% of the maximum density (see Figure 3).

Segments shorter than 10 s are considered undetectable, while segments between 10 and 45 s are partially detectable. Only segments exceeding 45 s are classified as fully detected.

Pseudo Code Ocular Exercises SearchInput:Video FileTranscript DictionaryProcess: 1.Identify Ocular Phase (using Transcript Dictionary): ○Search for the end of the “Questionnaire” section in the transcript dictionary.○Search for the start of the “Cheek Puff” exercise in the transcript dictionary.○Return the start and end timestamps of the identified ocular phase window.2.Process Video Frames (using Computer Vision):
○For each frame in the video,■If no face is detected:■Return “No Value”■If a face is detected: ■Use Dlib to place landmarks on facial features.■Crop the frame to the region containing both eyes based on the landmarks.■Classify the cropped eye region using the CNN.■If the classification confidence is greater than 0.95,■Record the classified gaze direction (up, left, right, or neutral).■Otherwise: ■Record “No Value”
○Return a list of gaze directions and their corresponding timestamps.3.Analyze Gaze Directions:
○For each gaze direction in the list,■Calculate a 5-s rolling window density curve for that direction.■Identify the timestamps of the longest segment in the density curve exceeding 60% of the maximum density.○Return a list of significant timestamps corresponding to the detected exercises.
Output: Ptosis timestamps (identified as the longest “up” gaze segment)Diplopia Right timestamps (identified as the longest “right” gaze segment)Diplopia Left timestamps (identified as the longest “left” gaze segment)

We can now remove from the video segmentation search in addition to the counting and cheek exercises, both in the ocular examination section and speech section. We are left with full body exercises, which test the strength of the arms and legs. In principle, the remaining section of the video can be analyzed using a body model.

### 2.7. Full Body Exercises

This section focuses on segmenting the two exercises within the limb strength test: arm extension and sit-to-stand.

We leverage the MediaPipe BlazePose body landmark detector [18] (see Figure 4) to identify relevant body parts in each video frame. This detector provides confidence scores for each identified landmark, indicating the certainty of the detection. The easiest of the two exercises to identify is the arm raise.

Focusing on the arm extension test, we restrict the analysis to video segments where at least the patient’s arms and shoulders are visible (refer to the pseudocode below). This ensures the effectiveness of body pose detection. The NLP results from the previous steps help define a search window to limit the processing time. We use the end of the first counting exercise as the beginning of this window. We track the positions and visibility of both shoulders (landmarks 11 and 13) and elbows (landmarks 12 and 14) within the identified body pose data. Using these landmarks, we calculate the angle between the upper arm (humerus) and the line connecting the shoulders (see Figure 5). At most, the patient maintains their arms extended for up to two minutes. While downward arm drifts are common in patients with MG, we use a minimum threshold of pi/4 for the humerus angle to determine when the arms are considered raised. This aligns with the clinical protocol used to identify the exercise completion. We identify the arm extension exercise as the longest time window in which both arms maintain an angle above the threshold. To ensure reliable detection, we analyze elbow visibility throughout the identified exercise window. If elbows are missing in more than 10% of the frames, the detection is labeled as uncertain.

Segmenting the sit-to-stand exercise presents a greater challenge compared to arm extension (we will discuss the reasons later). Here is our approach: we utilize the MediaPipe body pose detector to track the patient’s head position throughout the exercise window. We calculate the time derivative (gradient) of the head’s vertical position during the identified window. Sit-to-stand occurrences are identified as positive peaks in this gradient, signifying head movement upwards. Since patients typically perform the sit-to-stand maneuver 1 to 3 times, we search for this number of gradient peaks with similar characteristics in the time series. The exercise timestamps are then defined as the time points immediately before and after each detected peak where the head position remains stable, indicating standing and sitting postures.

Pseudo Code: Arm Raise SearchInput:Video FileProcess: 1.Iterate Through Video Frames: ○For each frame in the video:■Body Detection:■If a body is not detected using MediaPipe, return “No Value”.■If a body is detected, proceed to the next steps.■Landmark Placement: Place landmarks on the body using MediaPipe.■Arm Visibility Metric: Record a metric representing arm visibility (e.g., percentage of arm area visible).■Armpit Angle Calculation: Calculate the angles of both armpits based on the landmarks.■Return Values: Return a list of arm visibility metrics, a list of timestamps corresponding to each frame, and a list of armpit angles.

2.Segment Identification: ○Longest Segment Search: Identify the longest continuous segment where■Arm visibility metric is greater than 0.9 (e.g., at least 90% of the arm is visible).■Armpit angle is greater than 45 degrees (indicating raised arms).■This step can be accomplished using techniques like dynamic programming or a sliding window.3.Exercise Timestamps: Return the timestamps corresponding to the beginning and end of the identified segment as the arm exercise timestamps.
Output:Arm Exercise Timestamps (start and end)

Pseudo Code: Sit-To-Stand SearchInput:Transcript DictionaryList of Validation Words (words confirming sit-to-stand execution)List of Explanation Words (words preceding sit-to-stand instructions)Process:Keyword Search in Transcript:○Search the transcript dictionary for occurrences of both Validation Words and Explanation Words.Start of Exercise:○Identify the start of the exercise as the first occurrence of two Validation Words within a 5-s window in the transcript.Segment Building with Chains○For each Explanation Word found in the transcript: ■Start a 20-s chain.■If any Validation Word is found within the current chain, extend the chain duration by 10 s.○The end of the segment is identified as the last word within the most recently extended chain.Output Timestamps: Return the timestamps corresponding to the start and end of the identified segment as the sit-to-stand exercise timestamps.Output:Sit-To-Stand Timestamps (start and end)

We have detailed the segmentation of the video into nine clips, starting from the end of the MG-ADL test. Segmentation success may vary for each clip and is categorized as successful, uncertain, or unavailable. After segmenting the video, we employ our existing software modules to calculate individual scores for each test, as detailed in [11,13]. Machine learning has enhanced the robustness of these scoring algorithms through the dataset referenced in [15]. This two-step process automatically generates a comprehensive MG-CE score report that can assist neurologists in their diagnosis: Clinician Review with Graphical User Interface (GUI).

We recommend that the neurologist review the video in accelerated mode (enabled by our preprocessing) to validate, annotate, or disregard our scoring results (see Figure 6).

To address this need, we propose a dedicated GUI with the following functionalities:-Efficient Video Review: The GUI allows rapid playback of the patient’s video, focusing solely on the test segments the patient performed.-Annotated Landmarks: The video is overlaid with graphical annotations highlighting relevant anatomical landmarks during each test. This provides visual cues for the neurologist to assess the scoring accuracy.-Error Identification: In scenarios where our algorithms encounter difficulties, such as tracking the iris or eyelid during the ptosis exercise, the annotated video will clearly indicate potential scoring errors. This allows the neurologist to quickly identify issues and make informed decisions.

The benefits of the GUI are several. By enabling rapid review with visual cues, the GUI significantly reduces the time that neurologists spend on manual scoring analysis. The ability to verify and potentially correct scoring results mitigates the risk associated with relying solely on AI. The transparent review process facilitated by the GUI can strengthen the case for potential FDA approval of the software as a medical device. While a standard MG-CE score examination typically takes around 20 min (see Figure 7), our GUI streamlines the process by:-Fast Forwarding Non-Exercise Segments: During periods when the patient is not actively performing exercises, the video playback speeds up by a factor of 10 to minimize review time.-Slow Motion for Exercises: During exercise segments, the video playback slows down significantly, allowing the neurologist to clearly see the annotated landmarks and confirm scoring accuracy. This balance ensures an efficient review while maintaining sufficient detail.

Considering the average duration for each MG-CE active test window (1 min for each ocular test, 2 min for arm extension, 1 min for counting, etc.), the total review time with the annotated video in the GUI is approximately 3 min. This represents a significant improvement, as it is roughly six times faster than the original video acquisition time.

We will now delve into the results obtained using our multichannel divide-and-conquer segmentation technique.

## 3. Results

Before delving into the detailed results, we acknowledge that some videos may be excluded from the analysis due to factors like poor patient home conditions or limitations in telehealth data acquisition (details in Appendix B). The overall scoring performance of our digital method after video segmentation has been previously reported in [14]. Here, we focus specifically on the ability of our algorithm to process videos and identify/segment individual tests within them. We evaluated our algorithm on two datasets:The MGNet Project: This dataset comprises 86 videos of neurological clinical examinations for 51 myasthenia gravis (MG) patients across five different centers, performed by eight certified neurologists.Control Subjects: To account for potential variability in clinical examinations, we also processed a dataset of 24 videos featuring healthy control subjects who underwent neurological examinations under strict protocol adherence [15].

Each video was manually annotated to obtain the ground truth timestamps of every exercise in the dataset. This two-pronged approach allows us to assess the algorithm’s robustness under different examination conditions.

### 3.1. Ocular Exercises

The ocular phase for the MGNet dataset can be segmented with great reliability owing to our NLP algorithm. Out of 101 videos, the only video where this time segment is undetected has an incorrect recording that starts in the middle of the first exercise, missing a good section of the MG-CE examination record.

Next, we tested the accuracy of the direction of gaze of the patient on fixed images of the ocular phase, as shown in Table 2. The frames were classified according to the timestamp at which they were recorded. The model has a success rate of about 97%.

This rate of detection is for individual frames, and the algorithm achieves better performance on a video sequence of 5 s—see Figure 3B. However, patients may not maintain their gaze in the proper direction during the ptosis and diplopia tests.

When the algorithm finds a test segment longer than 45 s, the score test is labeled as identifiable. If the algorithm finds a segment between 10 and 45 s, the score test is labeled as partially identifiable. If the algorithm provides no segments longer than 10 s, we claim that the score cannot be computed. Table 3 shows the results for the ocular exercises detectability.

Nevertheless, we found the start and end of each ocular test within 5 s on 87% of the video, and either the start or end of the exercise is detected within 10 s on 92% of the video. The time window is then accurate enough to obtain the score computed by our computer vision algorithm, as reported in [6,7].

This result improved considerably in control subjects who have less difficulty maintaining their gaze in the correct direction, as reported in Table 4.

The video segmentation function for all control subjects has satisfactory accuracy in order to run our score algorithm.

### 3.2. Speech Test

Next, we report the results of the speech tests in Table 5. Our criteria for success is based on the time difference between the start of the test and the NLP detection, which should be less than 10 s.

Failures occur when the patient is not loud enough or counts too fast for the transcription to be detected. However, the score should not be computed in these situations because either dysarthria or breathing capacity cannot be properly assessed.

The results are better for the control subjects, as shown in Table 6 below.

In one of the videos, the patient counted 50 for both tests; however, this should not be allowed in the protocol.

### 3.3. Cheek Puff and Tongue-to-Cheek

As discussed in [11,14], this test is not helpful from a neuromuscular point of view to evaluate patients with MG; there is a lot of variation in the anatomy of the face muscles among patients, and it is difficult for the doctor to judge the strength of those muscles without touching the patient, which is not feasible in telemedicine. We decided to group these two exercises into a single phase and discard that segment from the video analysis. Provided that the protocol is correctly implemented, our algorithm based on keywords in the transcripts works satisfactorily for that purpose but fails too often to obtain a robust evaluation of the start and time of each separated active test window for a score evaluation.

### 3.4. Arm Extension

The segmentation of the arm extension is set to identify the start of the test within 10 s. This result is satisfactory for the MGNet dataset because failures correspond only to very poor visibility of the elbows of the patient during video acquisition, which can be assessed automatically. In this situation, the score should not be computed as reported—in [11]. Table 7 and Table 8 show the success rates on patient and control videos respectively.

The results are better than those expected for the control subjects.

### 3.5. Sit-to-Stand

For sit-to-stand, the adherence to the protocol of the MGNet dataset is very poor, i.e., the video does not record a view of the patient from toe to head during the test. Often, the head and half of the body below the hip are missing. As reported in [11], the score should not be computed on these patients. On the contrary, the protocol was enforced on the control patient, and the algorithm worked correctly all the time. Out of the 24 videos, 23 were segmented properly, and our algorithm correctly recognized that one patient with a large BMI did not perform the exercise during the clinical examination.

## 4. Discussion

Telehealth provides a distinct opportunity to harness the power of digital technology and artificial intelligence. Because all communication occurs in a digital format, it can be easily stored, reviewed, and analyzed using advanced algorithms [19,20].

This study explored the development of a software solution for automated scoring of myasthenia gravis (MG) examinations using telemedicine videos. We utilized datasets from the MGNet project (51 patients with MG) and our own control subjects (24 healthy individuals). This combined dataset represents the current standard in clinical neuromuscular examinations for patients with MG. The manual effort needed to exploit the MGNet dataset was considerable, requiring human analysis of 50 h of video and 1700 pages of reports. This work was, however, necessary for validation purposes. By combining AI with digital processing, we aim to automate, improve efficiency, and standardize the process as we refine the technology.

We found [11] that while the overall MG-CE examination scores were consistent across examiners, individual metrics showed significant variability, with up to a 25% difference in scoring within the MG-CE range. This was particularly noticeable in the examinations of patients with low disease severity. Such variability can significantly hinder the ability to detect true treatment responses in clinical trials, especially in patients with milder symptoms. Our observation is consistent with prior studies on the reproducibility of neurological examinations in other neurological conditions [21,22]. Most often, clinical trials fail. There are many reasons for this, including poor assessment of the benefits for patients [23]. Unbiased digital data on symptoms should contribute to objectively assessing the benefit of a Myasthenia Gravis drug for patients.

Clinical examination to obtain an MG-CE score, for example, may not be performed by a certified neurologist. This role of performing clinical trial assessments is often delegated to research coordinators after a period of training with limited variation in mastery [7,8].

However, standard clinical examinations for MG are time-consuming, expensive, and infrequent, making it difficult to closely track patient conditions and potentially prevent MG crises.

We propose a solution that can automatically run uniformly at a central location of the patient dataset to provide consistent scoring and provide plenty of opportunities to revisit the dataset as needed.

Our approach should allow us to scale up clinical trials and lower their cost for the scoring parts. In return, one should have a larger dataset for a rare disease and have the opportunity to improve the digital algorithm, which is essentially a combination of machine learning techniques.

Our solution employs the following three-step process:Video Segmentation: Individual test segments are extracted from the raw clinical examination video.Automated Scoring: Specialized software computes the MG score for each segmented test.Clinician Review: A Graphical User Interface (GUI) facilitates clinician review of the results and validation of the scoring.

This study focused on the first step, video segmentation. We refer to our previous work for details on automated scoring using this software. The primary challenge in developing a robust digital solution is to achieve consistent and reliable performance across diverse conditions. Video quality variations and the inherent variability in neurologist examination practices pose significant hurdles. To address these issues, our algorithm outputs various feedback messages. Score calculation success or failure: This aids in identifying instances in which poor visibility or audio quality hinders accurate scoring. Data acquisition improvement suggestions: This feedback can be used to enhance future data collection procedures and the training of medical staff.

Automated scoring using our approach could potentially facilitate clinical trials by providing centralized and consistent MG score evaluation; our solution can streamline and reduce the cost of scoring in clinical trials for MG drugs. Larger datasets can be generated for rare diseases like MG, which can further improve the accuracy of the machine learning algorithms used in the scoring process.

The graphic user interface, however, should be carefully validated with a human factor study. While our solution was developed under the guidance of a few clinicians, it will require a systematic protocol with a larger group of users to test our GUI and make sure that there are no hidden factors that may lead to misinterpretation [24]. This work identified some limitations that can be addressed with larger datasets. We found that relying solely on computer vision, natural language processing (NLP), or deep learning models might be insufficient for robust video segmentation. We propose combining these techniques and incorporating additional mechanisms as data volumes increase. While our divide-and-conquer approach works well offline, it requires as an input the entire video at once. Future work will focus on developing real-time processing capabilities to address poor on-the-fly data acquisition. Large language models offer promising avenues for robust solutions due to their vast training datasets. However, ethical considerations and the limited availability of training data for neurological conditions remain challenges. For example, healthy subjects exhibit different gaze patterns than patients with MG. We believe that addressing these limitations using larger datasets and further research will ultimately lead to a more robust and reliable automated MG scoring solution.

## 5. Conclusions

The clinical examination methodology of myasthenia gravis patients is shared by a broad spectrum of neurological disorders. By integrating advanced computational techniques, we aimed to enhance the precision and objectivity of quantitative clinical assessments of patients. We believe that the lessons learned from our work will advance the field of computing and improve quantitative patient clinical assessments. We also hope that our digitally robust and quantitative assessment telemedicine system will improve patient care in low-resource areas with limited access to neurologists.

## Figures and Tables

**Figure 1 bioengineering-11-00942-f001:**
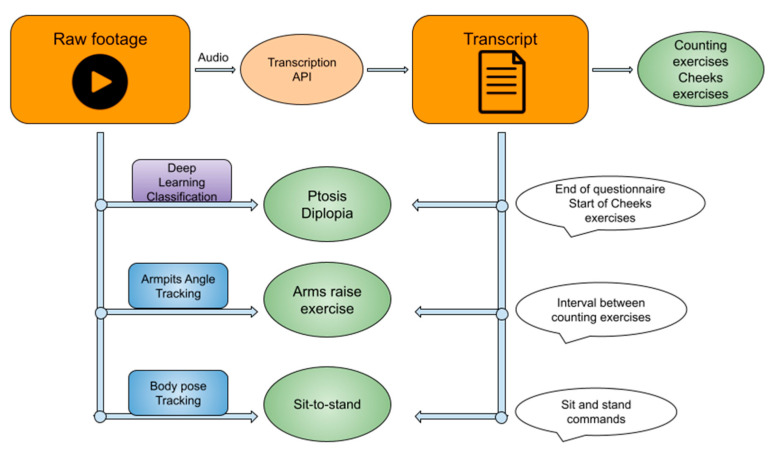
Overall process chart for separating the video clip.

**Figure 2 bioengineering-11-00942-f002:**
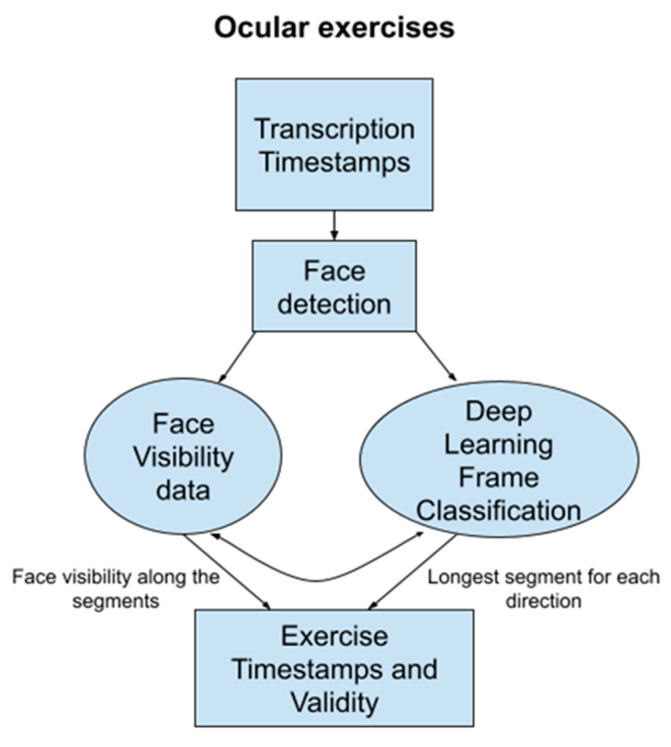
Algorithm to identify the time window for each ocular exercise to assess ptosis and diplopia.

**Figure 3 bioengineering-11-00942-f003:**
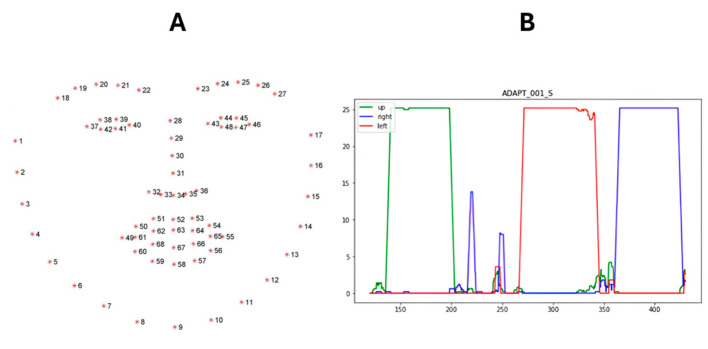
(**A**): Dlib 68 points landmark used to find the ROI of both eyes; (**B**): Detection of eye gaze looking successively up, then left, and right using a deep learning algorithm with the ROI. The curves represent the density for each gaze direction (up, left, and right) over the ocular exercise time window.

**Figure 4 bioengineering-11-00942-f004:**
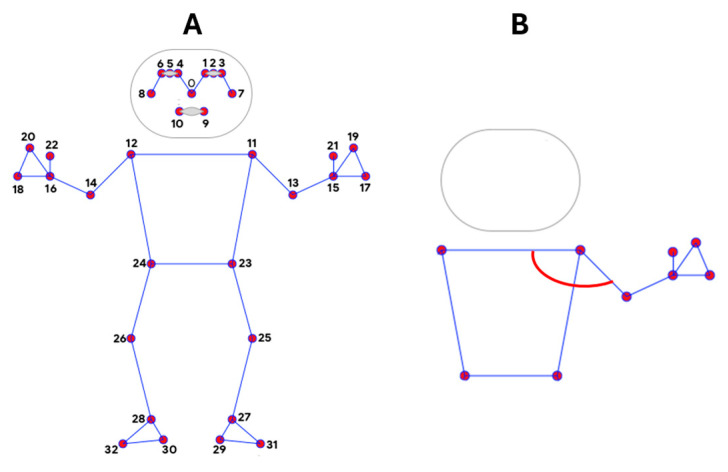
(**A**) Mediapipe landmark model used to detect body motion. (**B**) Representation of the angle tracked during arm raise.

**Figure 5 bioengineering-11-00942-f005:**
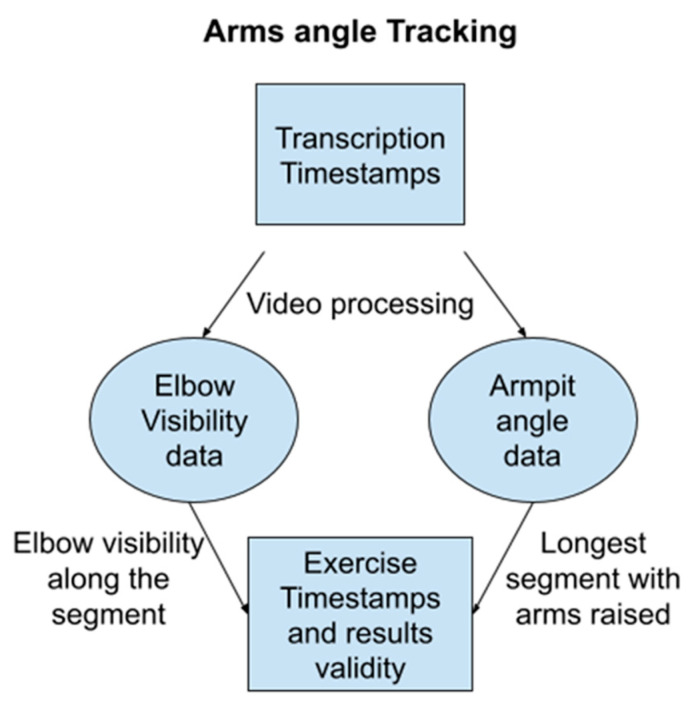
Algorithm to identify the time window of the arm strength exercise.

**Figure 6 bioengineering-11-00942-f006:**
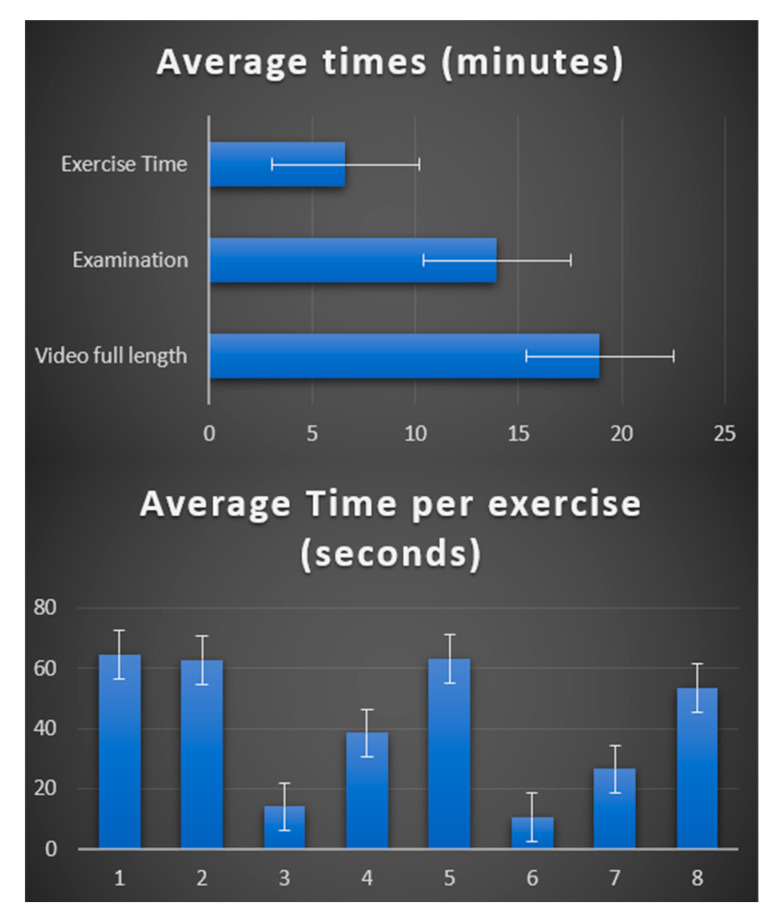
Average time distribution of video consultations.

**Figure 7 bioengineering-11-00942-f007:**
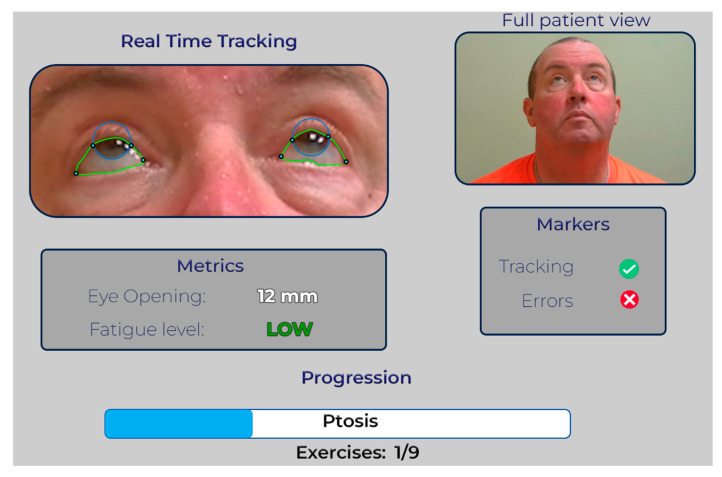
Doctor side GUI.

**Table 1 bioengineering-11-00942-t001:** MG-CE scores and criteria.

MG-CE Scores	Normal	Mild	Moderate	Severe	Item Score
Ptosis (61 s upgaze)	0 no ptosis	1 Eyelid above pupil	2 Eyelid at pupil	3 Eyelid below pupil	
Diplopia **	0 No diplopia with 61 s sustained gaze	1 Diplopia with 11–60 s sustained gaze	2 Diplopia 1–10 s but not immediate	3 Immediate diplopia with primary or lateral gaze	
Cheek puff	0 Normal seal	1 Transverse pucker	2 Opposes lips, but air escapes	3 Cannot oppose lips or puff cheeks	
Tongue-to-cheek **	0 Full convex deformity in cheeks	1 Partial convex deformity in cheeks	2 Able to move tongue-to-cheek but not deform	3 Unable to move tongue into cheek at all	
Counting to 50	0 No dysarthria at 50	1 Dysarthria at 30–49	2 Dysarthria at 10–29	3 Dysarthria at 1–9	
Arm strength ** (shoulder abduction)	0 No drift during >120 s	1 Drift 90–120 s	2 Drift 10–89 s	3 Drif 0–9 s	
Single breath count	0 SBC > 30	1 SBC 25–29	2 SBC 20–24	3 SBC < 20	
Sit-to-stand	0 Sit-to-stand with arms crossed, no difficulty	1 Slow/extra effort	2 Uses arms/hands	3 Unable to stand unassisted	
** When the test is performed on the right and left, score the more severe side				Total score: /24	

**Table 2 bioengineering-11-00942-t002:** Size of the training and testing sets.

Class	Up	Left	Right
Images in the training set	42,493	30,930	30,254
Images in the testing set	10,057	4289	5119

**Table 3 bioengineering-11-00942-t003:** Ocular test results for patients.

MGNet Ocular Test	Ptosis	Diplopia (Left)	Diplopia (Right)
**Total videos**	86	86	86
**No Score**	25	41	43
**Score Partially identifiable**	10	15	12
**Score Fully identifiable**	51	30	31

**Table 4 bioengineering-11-00942-t004:** Ocular test results for controls.

Control Ocular Test	Ptosis	Diplopia Left	Diplopia Right
**Total videos**	24	24	24
**No Score**	6	3	0
**Score Partially identifiable**	1	4	2
**Score Fully identifiable**	17	17	22

**Table 5 bioengineering-11-00942-t005:** Breathing exercise results for patients.

MGNet Test	Count to 50	Single Breath Count
**Total number of videos**	86	86
**Number of successes** **Difference < 10 s**	75	62

**Table 6 bioengineering-11-00942-t006:** Breathing exercise results in controls.

Control Test	Count to 50	Single Breath Count
**Total number of videos**	24	24
**Number of successes** **Difference < 10 s**	23	22

**Table 7 bioengineering-11-00942-t007:** Successes on patient videos.

**Total Number of Videos**	86
**Number of successes** **Difference < 10 s**	82

**Table 8 bioengineering-11-00942-t008:** Successes with control videos.

**Total Number of Videos**	24
**Number of successes** **Difference < 10 s**	23

## Data Availability

The data presented in this study are available upon request from the corresponding author for patient privacy.

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
