# Peer review of "AI-Powered Telemedicine for Automatic Scoring of Neuromuscular Examinations"

_bioengineering, 2024, doi:10.3390/bioengineering11090942_

Round 1
Reviewer 1 Report
Comments and Suggestions for Authors
The authors proposed a telemedicine framework for automatically scoring neuromuscular examinations. The following are the few comments that need to be addressed in the revised version of the paper.
1. Introduction: Describes the background by illustrating the general area of research. It also mentions the importance of the selected research area, which is crucial because [reason for importance]. Briefly overview the existing practices and limitations of the existing practices.
2. The research gap is not very clear.
3. Classify the existing methods into some categories. Define various attributes and compare the methods based on those attributes. Summarize the limitations of the existing methods highlighted and how your solution will address the shortcomings.
4. Figure 1 involves four significant technologies: audio-to-text API, Deep learning-based classification, Angle tracking, and body pose tracking. The paper does not explain how these tasks are achieved, including technical details of the methods and individual performance analysis.
5. Page 5: “Cheek puff and tongue to check the push…” How is muscle weakness detected in the video?
6. Segmentation related to each task and its accuracy should be discussed. Furthermore, how are the segmented areas tracked in the videos?
7. What NLP techniques are involved in MG-CE tests within the transcript?
8. Page 6: “Some video sections are easy to detect with NLP.” How?
9. Figure 2: What methods are used in face detection, frame classification, validity, etc.
10. The methodology section should adequately explain all the methods based on figures.
11. Automated test scores should be compared with the results of the human examiners, and a simplified comparison should be included.
Comments on the Quality of English Language
Proof-reading is required.
Author Response
The authors proposed a telemedicine framework for automatically scoring neuromuscular examinations. The following are the few comments that need to be addressed in the revised version of the paper.
- Introduction: Describes the background by illustrating the general area of research. It also mentions the importance of the selected research area, which is crucial because [reason for importance]. Briefly overview the existing practices and limitations of the existing practices.
Answer: we have rewritten the introduction trying to address your point and more.
- The research gap is not very clear.
Answer: we have done in our previous work [6] [7] [9] [13] the standard academic work that consist to describe the new method for digital scoring [6,7], extensive validation in [13] and finally comparison with clinical work in [9] tp show that digitalization gives better results. This is however far from a practical solution required by clinician: it is standard for them to do a video call and get their patient feedback, but there are not keen on using a check list, or an ap, or a rigorous order of clinical examination: doctors like to have a natural friendly interaction/conversation with patients to build their opinion. How do we do from a standard zoom call to a fully automatic report documentation is what we have attempted to show.
- Classify the existing methods into some categories. Define various attributes and compare the methods based on those attributes. Summarize the limitations of the existing methods highlighted and how your solution will address the shortcomings.
Answer: We have been improving the presentation to address your points, especially in the discussion section.
- Figure 1 involves four significant technologies: audio-to-text API, Deep learning-based classification, Angle tracking, and body pose tracking. The paper does not explain how these tasks are achieved, including technical details of the methods and individual performance analysis.
Answer: we also use a mixt of deep learning and classical segmentation technique for vision scores [7], various NLP for processing the text, and many more technologies. I suggest that the main interest of this paper is to demonstrate that those techniques can be used to solve a real clinical problem.
The new problem, we had to solve was video segmentation into clips corresponding to each test with a set of relatively poor quality video call acquired in true clinical conditions by neurologist, (not engineers). The main difficulty was to combine synergistically many existing methods to solve that problem. We have been referring to existing libraries as precisely as we could and refer to [6,7,13] for verification and validation of these libraries or own algorithm (as well as the literature on these libraries).
.
- Page 5: “Cheek puff and tongue to check the push…” How is muscle weakness detected in the video?
Answer: this is an excellent question. In principle the clinician detect visually the leak of air and cheek deformation. It runs out that this is not working well as shown in our human factor study. This exercise should be removed from the MG-CE score, but it is not our decision.
- Segmentation related to each task and its accuracy should be discussed. Furthermore, how are the segmented areas tracked in the videos?
Answer: Indeed! We have provided in the result section systematically the success rate of each video clip segmentation.
We process manually 50 hours of video and checked 1700 pages of report for the MGnet study as well as about 12 hours of video from control subjects. This provided the ground true for our error estimate. This is a very tedious process as mentioned in the discussion section.
- What NLP techniques are involved in MG-CE tests within the transcript?
Answer: we have details on the NLP in the method section and appendix our techniques. We start with simple things like identifying key words and eventually had to ramp up with expression and combination of words. As mentioned all along the paper not a single method works in isolation: video quality is poor, transcript can be faulty, protocol followed by the clinician varies: we have shown in our pseudo algorithm how all these techniques can be combined synergistically to provide the result.
- Page 6: “Some video sections are easy to detect with NLP.” How?
Answer: The video sections containing a fixed script, like the ADL questionnaire and the counting exercises are the easily detected parts. This clarification was added to the text.
- Figure 2: What methods are used in face detection, frame classification, validity, etc.
Answer: we have set references in the text to each of the libraries we used, including for face detection, body position ect…
- The methodology section should adequately explain all the methods based on figures.
Answer: we agree but since we use a combination of exiting methods well documented and referenced in this paper, we have given pseudo algorithm to show how these libraries can be combined.
- Automated test scores should be compared with the results of the human examiners, and a simplified comparison should be included.
Answer: yes indeed. This was done in our prelim work – see [9] and [13] in particular.

Reviewer 2 Report
Comments and Suggestions for Authors
This paper focuses on a particular aspect of an automatic scoring of Neuromuscular Examinations, i.e., the inclusion of MLP tools for aiding video segmentation, a required preprocessing step to identify the appropriate segments of a video where the concrete actions to evaluate occur. The paper is well-written, very easy to follow, and the methodology, data acquisition protocols, and ethical aspects are sound.
The only concern is that the use of MLP for aiding video segmentation is of course not novel, although its inclusion in this particular toolchain is. However, the extent of the contribution in this work is clearly and honestly stated.
As a minor remark, at the end of the second paragraph of the introduction the authors mention that "Our technology offers the potential to reduce costs and increase accuracy through the following methods: 1), ..., 4) ..." But the 1) to 4) are not description of methods. This needs to be rephrased.
Author Response
Reviewer 2
This paper focuses on a particular aspect of an automatic scoring of Neuromuscular Examinations, i.e., the inclusion of MLP tools for aiding video segmentation, a required preprocessing step to identify the appropriate segments of a video where the concrete actions to evaluate occur. The paper is well-written, very easy to follow, and the methodology, data acquisition protocols, and ethical aspects are sound.
The only concern is that the use of MLP for aiding video segmentation is of course not novel, although its inclusion in this particular tool chain is. However, the extent of the contribution in this work is clearly and honestly stated.
Answer: thanks for this comment. It is very fair. While we spend significant time in our previous work, to build the digital method for scoring, validate it on a large data set and comparing our digital score to score provided by neurologist [6,7,9,13], we found that getting a code that does the job fully automatically from a standard zoom video call to the report editing is far from trivial. Some time the last mile is the most difficult one. Clinical acceptation of our tool will still be one order of magnitude even more difficult.
As a minor remark, at the end of the second paragraph of the introduction the authors mention that "Our technology offers the potential to reduce costs and increase accuracy through the following methods: 1), ..., 4) ..." But the 1) to 4) are not description of methods. This needs to be rephrased.
Answer: absolutely. We list our rational following a certain logic and added references to sustain it.

Reviewer 3 Report
Comments and Suggestions for Authors
This article presents an automated telemedicine tool for the assessment of neurological conditions in patients with myasthenia gravis. Although the basic idea is interesting and potentially useful, the work has several shortcomings in terms of clarity of presentation and consistency.
First of all, the description of the pathology and the clinical tests used (MG-ADL, MG-CE) is insufficient, making it difficult for the reader to fully understand the context and objective of the study. Furthermore, the logical flow of the introduction is confusing.
Regarding the methods, there are several unclear points and the results of the video segmentation are not satisfactorily presented.
By points:
- the first paragraph of the introduction from line 30 to 35 is unclear, first it says why we want to use telemedicine for MG then what MG is.
- it is said that the measures used to date are not reproducible but this sentence is not supported by reference.
- also at line 56 individual scores are mentioned but not explained, it is not clear what is being referred to.
- the chapter on materials and methods should begin by explaining well what you want to do, it is not clearly written.
- why is the dataset explained on line 64 and then again afterwards? moreover it is not said how it is composed, except for the normal controls, is it an existing DB or was it created for this work?
- line 65 MG composite examination has the same acronym as MG core examination?
- line 67 MG-CE is an error and should be MG-ADL?
- The novelty proposed in this work is the segmentation algorithm, not the entire workflow, and this must be clear from the beginning.
- Tables and figures are sometimes capitalised, sometimes lower case, sometimes the caption of the table is below the table sometimes above.
- the chapter on the segmentation method is confusing and not very technical.
- at line 205 why is the test description in brackets when it should be fundamental information?
- at line 211 ‘we can analyse’ so has it been analysed or not?
- also on line 219 there is confusion because the exercises are not well explained beforehand.
- some paragraph titles have ‘:’ others do not.
- line 322 the word ‘typically’ is very vague.
- line 326 the word ‘ideally’ is used without specifying whether it is actually so, and what can be done otherwise.
- the results of segmentation are not clear, it is not clear whether a ground truth has been created.
- from line 432 there is a font variation and also at line 451.
- line 453 reexplains the dataset, and for the second time roughly.
- line 464 the videos are 101 or 102?
- are the tests named on line 467 actually done in another job?
- it is unclear how normal controls are used and what they are intended to prove.
- the discussion paragraph is useless: until line 561, things said in the introduction are repeated again, and again there is confusion about the actual contribution of the work.
- the discussion paragraph does not discuss the results, which are still unclear.
- the style of the quotations seems to change from one to the other.
Author Response
Reviewer 3
This article presents an automated telemedicine tool for the assessment of neurological conditions in patients with myasthenia gravis. Although the basic idea is interesting and potentially useful, the work has several shortcomings in terms of clarity of presentation and consistency.
First of all, the description of the pathology and the clinical tests used (MG-ADL, MG-CE) is insufficient, making it difficult for the reader to fully understand the context and objective of the study. Furthermore, the logical flow of the introduction is confusing.
Regarding the methods, there are several unclear points and the results of the video segmentation are not satisfactorily presented.
By points:
- the first paragraph of the introduction from line 30 to 35 is unclear, first it says why we want to use telemedicine for MG then what MG is.
- it is said that the measures used to date are not reproducible but this sentence is not supported by reference.
- also at line 56 individual scores are mentioned but not explained, it is not clear what is being referred to.
- the chapter on materials and methods should begin by explaining well what you want to do, it is not clearly written.
Answer: Thanks for all these points; It is very useful ; we have rewritten the intro to keep up with the logic as suggested, gives more detail on the disease and add many references. This work follow up two other piece of work published today as preprint (MedRxiv) but submitted to neurology journals. [9] is a human factor study that gives a lot of details on how human factor greatly impact the quality of scores, [13] is a large validation study on the MGnet cohort. Clinical Journal in neurology takes a long time to be reviewed as opposed to this journal in bioengineering.
- why is the dataset explained on line 64 and then again afterwards? moreover it is not said how it is composed, except for the normal controls, is it an existing DB or was it created for this work?
Answer: MGnet was sponsored by NIH and was a large multisite clinical study involving Harvard, Duke, Univ. of Chicago, Yale and George Washington University. 52 patient is a large cohort for a rare disease like MG. We used this data set retrospectively. This study did not involved controle subject. We add for this paper 24 control subject, to increase the quality of data set (we had a better quality control on the zoom video call, and rigor on protocol) and get a more diverse population. We described a first time the data set at the beginning of the section on method to provide the material used for this study. We described eventually the data set again with detail points as needed for the comprehension of the method and results.
- line 65 MG composite examination has the same acronym as MG core examination?
Answer: This error was fixed. MG-CE only refers to the Core examination
- line 67 MG-CE is an error and should be MG-ADL?
Answer: We have only two score the composite MG-CE that is given by the clinician rating symptoms at the end of each test, and MG-ADL that is noted as the answer of the patients to specific questions. It was not an error but there was no transition between the two phases of the examination. A sentence was added to clarify.
- The novelty proposed in this work is the segmentation algorithm, not the entire workflow, and this must be clear from the beginning.
Answer: This is now clearly stated in the introduction:
“We have constructed the computational framework in [6] and [7] to assess MG-CE scores digitally. This paper proposes a further step to make the MG-CE examination run fully automatically by identifying each part of the video examination and segmenting the video. A simple graphic user interface can run the annotated video of the patient evaluation produced by our system in fast mode to allow a certified neurologist to review the analysis and validate the score promptly, therefore speeding up the process by one order of magnitude.”
- Tables and figures are sometimes capitalised, sometimes lower case, sometimes the caption of the table is below the table sometimes above.
Answer: Table captions are now all placed above the tables.
- the chapter on the segmentation method is confusing and not very technical.
Answer: we have provided the pseudo algorithm for each part of the segmentation and references to each library that were used. We felt that doing the segmentation of the video in clip does not require to reinvent the wheel but rather require the proper combination of tools applied in the correct order (divide and conquer approach – as described in the paper). It took a considerable amount of tedious work to get a robust method with such a diverse data set of patients.
New algorithms are required for segmentation to get the score itself as described in our previous work [6,7].
- at line 205 why is the test description in brackets when it should be fundamental information?
Answer: Agree, thanks, the parentheses were removed and the content was integrated to the sentence.
- at line 211 ‘we can analyse’ so has it been analysed or not?
Answer: yes indeed: They have been analysed, ‘can’ was removed
- also on line 219 there is confusion because the exercises are not well explained beforehand.
Answer: Detected exercises are discarded from the timeline, making detection of subsequent exercises easier. This is a divide and conquer method that start solving the simplest problem, and keep moving on until the all video has been segmented into video clip.
- some paragraph titles have ‘:’ others do not.
Answer: Removed the ‘:’ for consistency across paragraphs
- line 322 the word ‘typically’ is very vague.
Answer: Removed typically, which was unnecessary and brought confusion
- line 326 the word ‘ideally’ is used without specifying whether it is actually so, and what can be done otherwise.
Answer: Replaced ideally with ‘at most’ to convey the idea more accurately
- the results of segmentation are not clear, it is not clear whether a ground truth has been created.
Answer: A ground truth was created by manually annotating the videos. This is now clearly stated in the text.
- from line 432 there is a font variation and also at line 451.
Fixed
- line 453 reexplains the dataset, and for the second time roughly.
- line 464 the videos are 101 or 102?
Answer: There are 101, this was corrected.
- are the tests named on line 467 actually done in another job?
Answer: The model was tested for this job. This is now clarified in the text.
- it is unclear how normal controls are used and what they are intended to prove.
Answer: we have clarified this in the description of the data set as requested, and also shown clearly in the result section the benefit of having that new data set because quality of video and protocol were done better than with MGnet.
- the discussion paragraph is useless: until line 561, things said in the introduction are repeated again, and again there is confusion about the actual contribution of the work.
- the discussion paragraph does not discuss the results, which are still unclear.
- the style of the quotations seems to change from one to the other.
Answer: Sorry about that. We have hopefully improved much the discussion section while keeping it short.

Reviewer 4 Report
Comments and Suggestions for Authors
The present work is very interesting as it deals with a major disadvandage of Telemedicine applications as human subjectivity limits the reliability of all medical testing i.e. for disability status. Authors could consider the following points to improve their work:
1. Introduction
lines 28-29. This sentence need more details based on more references.
lines 36-37: some references are needed
lines 44-45: need more details ans references
lines58-60: more details are needed
2. Discussion
lines 532-536: this paragraph need rephrazing in accordance of the title of the paper.
lines 565-568: this part of the discussion could be the stronger of this section identifiing the solutions of the weaknesses of the presented approach.
Conclusions
lines 594-595: This sentence should be more explained
Author Response
Reviewer 4
The present work is very interesting as it deals with a major disadvandage of Telemedicine applications as human subjectivity limits the reliability of all medical testing i.e. for disability status. Authors could consider the following points to improve their work:
- Introduction
Answer: Thank you, we have rewritten most of the introduction as suggested. Thereafter are some specific changes we did to answer your point.
lines 28-29. This sentence need more details based on more references.
Statement: “There is great promise in taking advantage of the video environment to provide re-mote alternatives to physiological testing and disability assessment”
lines 36-37: some references are needed
Statement: “Clinical trial outcome measures for many neurological diseases are compromised by subjectivity, poor reproducibility across evaluators, and need for in person evaluations”
lines 44-45: need more details ans references
Statement: “We also are able to capture deficits that are better detected by our technology than even expert human assessment”
lines58-60: more details are needed
Statement: “. A simple graphic user interface can run the annotated video of the patient evaluation produced by our system in fast mode to allow a certified neurologist to review the analysis and validate the score promptly, therefore speeding up the process by one or-der of magnitude.”
- Discussion
lines 532-536: this paragraph need rephrazing in accordance of the title of the paper.
lines 565-568: this part of the discussion could be the stronger of this section identifiing the solutions of the weaknesses of the presented approach.
Answer: absolutely. We had to extend and improve significantly the discussion section.

Round 2
Reviewer 1 Report
Comments and Suggestions for Authors
Authors have addressed the comments in the revised version.
Comments on the Quality of English LanguageMinor editing is required
Reviewer 2 Report
Comments and Suggestions for Authors
The comments raised in the previous review were addressed. I have no further comments.
Reviewer 3 Report
Comments and Suggestions for Authors
The authors have replied to the comments. Please note that the acronym MG is defined twice in the introduction.
Reviewer 4 Report
Comments and Suggestions for Authors
After these modifications, the manuscript is improved and ready to be published as it is.
Excellent work and well presented.